# The Low-Fluence Q-Switched Nd:YAG Laser Treatment for Melasma: A Systematic Review

**DOI:** 10.3390/medicina58070936

**Published:** 2022-07-14

**Authors:** Yeon Seok Lee, Yu Jin Lee, Jung Min Lee, Tae Young Han, June Hyunkyung Lee, Jae Eun Choi

**Affiliations:** Department of Dermatology, Nowon Eulji Medical Center, Eulji University School of Medicine, Seoul 01830, Korea; yeonseok211@gmail.com (Y.S.L.); sara941222@gmail.com (Y.J.L.); jml9641@gmail.com (J.M.L.); dermahan@gmail.com (T.Y.H.); drhams77@naver.com (J.H.L.)

**Keywords:** laser, laser toning, melasma, Q-switched Nd:YAG laser

## Abstract

Melasma is a common pigmentary disorder with a complex pathogenesis, of which the treatment is challenging. Conventional treatment often leads to inconsistent results with unexpected pigmentary side effects and high recurrence rates. Recently, the low-fluence Q-switched Nd:YAG laser (LFQSNY) has been widely used for treating melasma, especially in Asia. We reviewed literatures on the LFQSNY treatment of melasma published between 2009 and May 2022 to evaluate the efficacy and adverse events, including its combination therapy. A systematic PubMed search was conducted and a total of 42 articles were included in this study. It was hard to summarize the heterogenous studies, but LFQSNY appeared to be a generally effective and safe treatment for melasma considering the results of previous conventional therapies. However, mottled hypopigmentation has been occasionally reported to develop and persist as an adverse event of LFQSNY, which may be associated with the high accumulated laser energy. When used aggressively, even LFQSNY can induce hyperpigmentation via unwanted inflammation, especially in darker skin. Although few studies have reported considerable recurrence rates three months after treatment, unfortunately, there is a lack of the long-term follow-up results of LFQSNY in melasma. To enhance the effectiveness and reduce the adverse events, LFQSNY has been used in combination with other treatment modalities in melasma, including topical bleaching agents, oral tranexamic acid, chemical peeling, or diverse energy-based devices, which generally reduced side effects with or without significant superior efficacy compared to LFQSNY alone.

## 1. Introduction

Melasma is a commonly acquired pigmentary skin disease, most observed in adult females and darker skin types of Fitzpatrick phototypes III-V. Clinically, it presents as symmetrical ill-defined hyperpigmented patches on the face, often causing cosmetically serious psychosocial burdens to patients. The pathogenesis of melasma has not been fully understood; however, several factors including chronic ultraviolet exposure, hormonal changes (pregnancy and oral contraceptives) and genetic backgrounds have been proposed to play a role [1]. Treatment of melasma is one of the most challenging fields to dermatologists. The results are inconsistent and unsatisfactory; recurrence and even worsening of the condition during or after treatment is not uncommon. The classic standard treatment is the topical application of modified Kligman’s triple combination (TC), consisting of hydroquinone (HQ) 4%, tretinoin 0.05%, and fluocinolone acetonide 0.01%. Laser treatment has been relatively contraindicated for melasma due to the risk of inducing inflammation and stimulating melanogenesis through unwanted photothermal effects, especially in darker skin [2,3,4]. The reason for such treatment resistance is not yet understood, but the complex pathogenesis of melasma might be involved. The accumulated knowledge to date has suggested melasma as a complex photoaging disorder rather than a simple pigmentary disease. It is histologically characterized by the features of photoaging or dermal inflammation, in addition to active melanocytes, solar elastosis, increased dermal vascularization, increased mast cell count, and altered basement membrane [1,5,6,7]. This implies that the excessive thermal damage of conventional laser irradiation can stimulate inflammatory change by basement membrane disruption and cell apoptosis, resulting in clinical aggravation of melasma.

However, since the 2000s, the low-fluence Q-switched Nd:YAG laser (LFQSNY), commonly referred to as ‘laser toning (LT)’, has been accepted as a new gold standard of melasma treatment in Asia, where there is high demand for treatment. This technique involves multiple sessions (usually around 10 sessions) of weekly or biweekly 1064 nm QSNY treatment with a low fluence (usually 1–3 J/cm^2^), a collimated beam with a large spot size, and a frequency of 5–10 Hz. The endpoint of the procedure would be faint erythema. LT is known to selectively destroy melanin in melanophores, whereas melanin-containing cells are left undamaged, resulting in safe depigmentation of melasma [8,9]. In addition, one of the key advantages is that there is no downtime affecting patients’ daily lives since the epidermis remains intact. Instead, rather marginal outcomes and questionable long-term results considering many treatment sessions of 1–2 week-intervals are drawbacks of LFQSNY in melasma. To achieve and maintain better clinical results safely, a combination of LFQSNY with various other treatment modalities are commonly used in clinical practice [10,11]. The aim of this review article is to evaluate the overall efficacy, adverse events, and recurrence rates of LFQSNY for melasma. Moreover, we aim to assess the various combination therapy of LFQSNY in melasma.

## 2. Materials and Methods

A systematic review of literatures was conducted following the Preferred Reporting Items for Systematic Reviews and Meta Analyses (PRISMA) guidelines. We searched English literature on PubMed using the terms, “Q-switched Nd:YAG laser” or “laser toning” and ”melasma” within the period of 2009–2022. The last search was run on 1 May 2022. The inclusion criteria were any original articles with a clinical study evaluating melasma treatment using LFQSNY, not limited to the prospective, randomized, controlled trials (RCTs). Articles regarding combination treatment (LFQSNY with other treatment modalities) or modified solitary LFQSNY were included. Studies with too small of a sample size (n < 10), those materially difficult to reproduce, or inconsistent laser delivery within a non-randomized trial arm were excluded (Figure 1). Two independent investigators performed extraction of articles according to the criteria. We also manually checked the relevant references of the included literatures to prevent any missing data. Discussion was maintained until the two review authors agreed to accept the settlement. Data encompassing study design, patient and treatment characteristics, melasma type, efficacy outcomes, adverse events, and recurrence rates were summarized (Table 1). Since many studies have used heterogenous outcome measures to assess efficacy, we have tried to include numerous scoring systems such as physician-assessed quantitative and qualitative evaluation, and patient-oriented self-evaluation (Table 2). Sunscreen application was not mentioned in the table because all patients used sunscreen as part of their routine melasma management.

## 3. Results

A total of 125 articles were initially identified in the literature search, of which 43 were duplicates and 33 did not meet the inclusion criteria and were thus removed. In the retrieved 46 articles, 4 articles were additionally excluded according to the criteria. A total of 42 articles were finally included: 19 RCTs, 15 non-RCTs (10 single-arm trials, 5 controlled trials), and 8 retrospective studies (Figure 1). A total of 1736 melasma patients were included, whose Fitzpatrick’s skin types consisted of mostly type II-V. Parameters used for LFQSNY varied among studies. The most commonly used spot size was 6–10 mm with a fluence of 0.5–3.8 J/cm^2^. The number of passes varied from 1 to 10 passes, being most performed in less than 5 passes. Laser treatment sessions were performed usually at intervals of 1–2 weeks with the exception of several studies (especially, studies regarding combination therapy) at intervals of 4 weeks [16,33,37,41]. The time at which efficacy was first evaluated also differed between studies: during the study period, immediately at the end of treatment, 1 to 2 weeks post-treatment, or 1 to 3 months post-treatment. Therefore, direct comparison between heterogeneous studies was difficult, even between those that had the same outcome measures. In addition, validating methods for efficacy differed according to authors. As a subjective outcome measure, the melasma area severity index (MASI) or modified MASI (mMASI) has been frequently adopted. In terms of objective outcome measures, the melanin index (MI) and erythema index (EI) measured from Mexameter^®^ have been frequently used, whereas a few old studies used lightness index (L*I), relative lightness index (RL*I), and color difference (ΔE*ab) using a spectrophotometer. Physician’s Global Assessment (PhGA) and subjective patient satisfaction or patients’ global assessment were also widely adopted indices. The results of the literatures are given in Table 1.

### 3.1. Low-Fluence Q-Switched Nd:YAG Laser in Melasma

As a monotherapy, LT was performed in 5–15 sessions (usually 9–10) in most studies, showing favorable outcomes not only in subjective measures but also in objective measures [12,21,24,27,28,35,43,44,48,52,53]. Transient erythema and edema were most reported immediately after treatment. Rare, but serious adverse events included pigmentary side effects, such as mottled hypopigmentation (MH) and rebound hyperpigmentation (RH), which were more frequent in darker skin (Fitzpatrick skin type IV, V) [21,44,49,53]. However, some authors have reported aggravation or relapse of melasma three months after cessation of treatment [18,28,33,43,48,49,51,53]. EI scores were higher in refractory melasma [24].

There are few long-term studies of LFQSNY in melasma, whereas in most clinical trials the patients were followed up 1–3 months after completion of treatment. Gokalp et al. in a retrospective trial reported that relapse was observed in 20 out of 34 patients at a 1 year follow-up after a median of 8 sessions of LFQSNY [27]. Three months follow-up results have been considerably reported in literatures showing various recurrence rates. Dev T. et al. and Wattanakrai et al. described that melasma recurred in all their patients who were followed up for 3 months [18,49], whereas Hofbauer et al. and Zhou et al. mentioned 81% and 64% recurrence rates, respectively [28,43]. However, contrarily, many other authors have reported significant improvement at 3 months after treatment [15,17,39,41,44].

In comparison with other treatment for melasma, there was no significant difference in terms of efficacy and adverse events between LFQSNY and low-fluence 755 nm Q-switched alexandrite laser (QSAL). However, QSAL required much fewer passes than QSNY to reach the end point (1–2 vs. up to 8 passes), which is associated with the higher level of melanin absorption of 755 nm wavelength compared to 1064 nm [34]. Compared to LFQSNY, 532 nm QSNY did not significantly reduce MASI score. Moreover, MH and PIH were more frequently observed in the 532 nm QSNY group compared to the LFQSNY group (27.5% vs. 4.8%) [46]. Compared to the glycolic acid (GA) peel, LFQSNY significantly reduced MASI score. Severe adverse events were rarely reported in both treatment groups [46]. In a study comparing LFQSNY and topical silymarin cream, there was no significant difference in the reduction in mMASI score and the incidence of adverse effects in both groups [14]. Dev T. et al. reported that there was no significant difference in the reduction in mMASI, MI score, and subjective evaluation of patients between LFQSNY and TC cream [18]. LFQSNY showed significant reductions in the RL*I and mMASI score and favorable patients’ satisfaction compared to 2% HQ cream [49].

Certain studies tried to find the differences in efficacy and adverse events according to pulse duration or pulse delivery [13,26,33]. Comparing QSNY and picosecond Nd:YAG laser (PSNY) in terms of the efficacy, there was no significant difference between both modalities [13]. Dual-pulsed QSNY, which is also known as PTP (photoacoustic twin pulse) mode, was noninferior to single-pulsed QSNY in terms of efficacy with significantly less pain [26]. 

### 3.2. Combination of Low-Fluence Q-Switched Nd:YAG Laser with Other Energy-Based Device

The combination therapy of LFQSNY and other energy-based devices (EBD) showed better or similar efficacy with fewer adverse events compared to monotherapy [15,19,20,22,25,30,38]. Compared to LFQSNY alone, LFQSNY combined with fractional CO2 laser did not show a significant difference in outcome measures such as mMASI score, MI/EI score (*p* > 0.05). However, the risk of MH was lower in combination therapy compared to monotherapy. [15] The combination therapy of LFQSNY and fractional Er:YAG laser (FEYL) showed significantly higher improvement in Visioface^®^ scores and MI/EI scores than monotherapy. No serious adverse events were reported in both groups. [25] LFQSNY and fractional Er:Glass laser (FEGL) combined therapy tended to show better results in mMASI score and patients’ self-assessment than LFQSNY alone, which was not statistically significant (*p* > 0.05). [39] Compared to dual-pulsed LFQSNY alone, its combination with fractional microneedling radiofrequency (FMR) showed significantly superior results in efficacy including MI/EI, PSI, and mMASI scores, as well as less adverse events such as MH and RH [19,20]. 

The combination of LFQSNY and pulsed dye laser (PDL) showed a significantly higher reduction in the MASI score compared to LFQSNY alone in the patients who had visible vasodilation on dermoscopy. However, there was no statistically significant difference between LFQSNY monotherapy and combination therapy in the patients without visible vasodilation on dermoscopy [22]. The combination of QSNY and long-pulsed Nd:YAG laser (LPNY) have treated melasma patients (including refractory melasma) without serious adverse events [31,47], showing a significantly greater reduction in mMASI score compared to LFQSNY monotherapy. MH and RH also occurred less in the combination therapy (1.1% vs. 14.1%) [30]. The combination of LFQSNY and intense pulsed light (IPL) was found to be an effective alternative for melasma treatment, showing a significant decrease in MASI score with few serious adverse events [32]. The regimen of IPL followed by LFQSNY maintenance was also effective for the treatment of melasma [40], showing significant reduction in mMASI and MI scores compared to monotherapy [38]. 

### 3.3. Combination of Low-Fluence Q-Switched Nd:YAG Laser with Non-EBD Therapy

The combination therapy of LFQSNY and 30% GA peel lowered the RL*I, mMASI, and MI scores significantly compared to the LFQSNY monotherapy. Although adverse events were rare in both therapies, PIH and MH occurred in patients with Fitzpatrick skin type V (13.3%) [29,45]. Compared with LFQSNY alone, LFQSNY and Jessner’s peel combination therapy tended to be more effective in the reduction in mMASI score or PhGA, which was not statistically significant. There were no serious adverse events reported in both groups [36]. The combination therapy of LFQSNY and modified Jessner’s solution peel did not show a significant difference in efficacy compared to LFQSNY monotherapy. However, the incidence of MH was lower in the combination therapy group compared to the monotherapy group (0% vs. 21.05%) [23]. Microdermabrasion and LFQSNY combination therapy has been proven to be effective in patients with refractory melasma [41]. 

In a study comparing LFQSNY and topical 3% tranexamic acid (TXA) gel versus microneedling and topical 3% TXA gel, there was no significant difference in reduction in mMASI score and patient satisfaction. However, this study had adopted very low fluence (0.8 J/cm^2^) with a spot size of 2.5 mm and 4 mm [17]. The combination therapy of LFQSNY and 20% azelaic acid cream showed significant improvement in MASI score compared to LFQSNY alone. However, there was no significant difference in efficacy between LFQSNY and 20% azelaic acid cream. In comparison with LFQSNY, a burning sensation was reported only in the 20% azelaic acid cream group (5%) [42]. In patients with refractory melasma who did not respond to HQ cream and TCC, LFQSNY and 7% alpha arbutin solution combination therapy showed clinical improvement [51].

LFQSNY and oral tranexamic acid combination therapy showed a significant decrease in mMASI score [17], and a significantly greater reduction in mMASI score compared to LFQSNY monotherapy [37]. 

## 4. Discussion

The exact action mechanism of LFQSNY in melasma has not yet been elucidated. Despite the number of the clinical study, there are few studies on the histopathologic and molecular study of melasma as there are few volunteers for skin biopsy due to cosmetic issues. However, based on a couple of studies, the selective destruction of the melanosomes with minimal thermal damage of melanocytes is considered to be the key concept of this technique, which is called ‘subcellular selective photothermolysis’ [8,9]. Using the zebrafish model in which the melanophores are externally visible, Kim et al. showed that at a certain low fluence, QSNY selectively photothermolyse melanosomes without killing melanocytes, whereas widespread apoptosis was observed at a higher fluence [9]. In an electron microscope study of human skin, the number of melanocyte dendrites were decreased, and stage IV melanosomes were selectively destroyed whereas early-stage melanosomes were unchanged after LFQSNY in melasma. As mature stage IV melanosomes are accumulated in the dendrites of melanocytes, it is assumed that the QSNY photothermolyse the mature melanosomes, leading to functionally downregulated melanocytes with fewer dendrites [8]. These findings were consistent with the histologic examination, demonstrating a reduced expression of melanogenic proteins (TRP-1, TRP-2, NGF, a-MSH and tyrosinase) as well as melanin (Fontana-Masson staining) in the lesional skin after LFQSNY, whereas the number of melanocytes (Melan-A and SOX-10) was unchanged after treatment, which reassured the concept of the subcellular selective photothermolysis [54].

### 4.1. Low-Fluence Q-Switched Nd:YAG Laser in Melasma

In our study, it was impossible to sum up the results of 42 heterogeneous studies. Nevertheless, most studies showed favorable results in both objective and subjective assessment as shown in Table 1. Meanwhile, there were a few studies reporting less effectiveness. Park et al. and Fabi et al. reported a 16.7% and 22% reduction in mMASI score, respectively, both after a total of six treatment sessions of weekly QSNY monotherapy, which might be associated with the insufficient total number of treatment sessions [34,45]. Interestingly, in a prospective, split-faced, randomized trial comparing LFQSNY and TC for 12 weeks, there was no significant difference in efficacy between the groups whereas adverse reactions were significantly more common in the TC side (erythema), which may imply overuse of TC. Nevertheless, it reminds us of the effective value of TC, the classic mainstay of melasma treatment.

Although LFQSNY is a relatively safe treatment for melasma by the aforementioned mechanisms, adverse events occasionally occur. Among them, MH or punctate leukoderma is the major concern since it lasts long without treatment. The incidence rate of MH is unknown. Although a larger portion of published studies have reported no or less incidence of MH, there are a couple of literatures reporting approximately a 10% risk of MH from LFQSNY in East Asian patients with melasma [49,55]. A retrospective analysis of a large number of 177 patients of melasma by Choi et al. demonstrated consistent findings that MH occurred in 21 out of 177 patients (11.9%) within 10 sessions of LFQSNY [30]. Although the underlying mechanism is not understood, the histopathologic exam shows a preserved number of melanocytes even in the MH lesion compared to the adjacent normal skin, which signifies that melanocytes still survive, but are functionally downregulated [54,56]. Intervention to stimulate melanogenesis in melanocytes using focused, narrow-band ultraviolet B therapy has been used with some success [57]. Although there are no statistical analyses, some authors have mentioned that hypopigmentation was generally sustained over 2–3 years, and spontaneous resolution was seen in only <10% of the patients after a 2-year follow-up [54]. Another report estimated that MH resolved in half of cases after 2 years and 80% after 3–4 years from their clinical experience [30]. The risk factor of MH is known to be the excessive cumulative energy; the use of relatively high fluence, short treatment intervals, and too many sessions of total treatment [54,58,59,60]. Therefore, caution is needed to avoid aggressive treatment and treatment should be discontinued as soon as possible upon the development of MH. 

There were few long-term follow-up studies on LFQSNY for melasma but several studies mentioned conflicting results three months after cessation of treatment. It may be associated with the difference in the individual lifestyle, including sun exposure, as well as the treatment settings and skin phototype. 

More recently, a novel PSNY system has been introduced as a new therapeutic option for pigmentation, which has an even shorter pulse duration of the picosecond (10^−12^) than the nanosecond (10^−9^) of QSNY. Theoretically, laser toning using PSNY is expected to have advantages over LFQSNY, since a picosecond laser can deliver a higher peak power effectively with much lower energy and less thermal damage to the surrounding tissue. However, in clinical practice, PSNY is not markedly superior to LFQSNY in melasma, yet it is still preferred in tattoo removal and acne scar treatment. Although there are only few reports, a split-face study comparing LFPSNY and LFQSNY for treating melasma demonstrated that neither was superior in pigment lightening [13]. However, the fractionated picosecond laser beam may enhance the efficacy and safety of melasma treatment by rejuvenating the dermal environment. It produces focal vacuoles in the epidermis and dermis by photomechanical effects, termed ‘laser-induced optical breakdown’, leading to dermal remodeling [61,62,63]. Further studies are needed on this novel laser system.

### 4.2. Combination Therapy of Low-Fluence Q-Switched Nd:YAG Laser in Melasma

The efficacy, tolerability, and adverse events of combination therapy of LFQSNY and other EBD or non-EBD were briefly summarized in the Table 3. Despite the combination of LFQSNY with fractional CO_2_ laser, FEYL, FMR, PDL (only in patients with visible vasodilatation on dermoscopy), LPNY, IPL, GA peeling, topical azelaic acid, or oral TXA showed significantly superior efficacy to LFQSNY alone, whereas FEGL and modified Jessner’s solution peel did not. However, it is notable that adverse events, such as MH and RH, were generally reduced in combination therapy (FEGL, FMR, LPNY, modified Jessner’s peeling) compared to LFQSNY alone [15,19,20,23,30]. Since melasma has a heterogeneous pathology as mentioned earlier, pigment-nonspecific treatment which targets the dermal pathology of melasma may exert synergistic effects by ameliorating the dermal environment. Moreover, fractional lasers can facilitate the transport and extrusion of epidermal melanin as well as dermal contents through the microscopic treatment zone, which is called the melanin shuttle function [39]. IPL can enhance the improvement of melasma using a distinct mechanism, different from the QSNY accelerating epidermal turnover. The processes, including the collapse of the melanin cap structures and melanosome concentration, are initiated after IPL irradiation forming an intraepidermal microcrust, which desquamates from the skin within 5–7 days. Although initial improvement is relatively dramatic in IPL, RH can also be frequently encountered as melanosomes are quickly replenished with reactivation of melanocytes. Thus, QSNY maintenance therapy may aid in stabilizing the improved state of melasma after IPL irradiation [40]. Although the results of the comparison are conflicting, it is meaningful that the serious pigmentary adverse events tend to be less frequent in combination therapies, which may be attributed to saving QSNY energy and stabilizing melasma lesions.

## 5. Limitations

Our study has several limitations. First, we did not limit our review to RCTs, but we also included retrospective and non-randomized trials; therefore, a potential bias could not be ruled out. Second, studies reviewed in this study had heterogenous designs with various sample sizes and outcome measures, thus it was difficult to compare the results head-to-head. Third, many studies were conducted over a short period of time. Long-term data regarding the recurrence rate and adverse events were limited. Fourth, we did not focus on the type of melasma. In future studies, a head-to-head comparison using a unified outcome measurement and a time point to evaluate the efficacy will be required, considering long-term data such as recurrence rate and type of melasma.

## 6. Conclusions

LFQSNY has become a preferred treatment of choice in melasma, in which traditional laser treatment is relatively contraindicated due to the high risk of post-treatment hyperpigmentation and high recurrence rate, especially in dark skin. Despite the unusual adverse events, such as MH, it is considered to be generally effective with minimal adverse events for melasma by selectively destroying melanosome while leaving melanin-containing cells intact. Excessive cumulated laser energy is known to be associated with the development of MH. There is a lack of long-term studies that follow patients post-treatment for longer than three months. Although it is conflicting, a few studies showed a high recurrence rate three months after cessation of LFQSNY. However, by using LFQSNY combined with other melasma treatment modalities, recurrence rates as well as adverse events can be reduced, with or without superior efficacy compared to LFQSNY alone. Since there is still no cure and long-term relapse may be inevitable, the importance of patient counselling on the relapsing course of melasma and the importance of photoprotection cannot be overemphasized. 

## Figures and Tables

**Figure 1 medicina-58-00936-f001:**
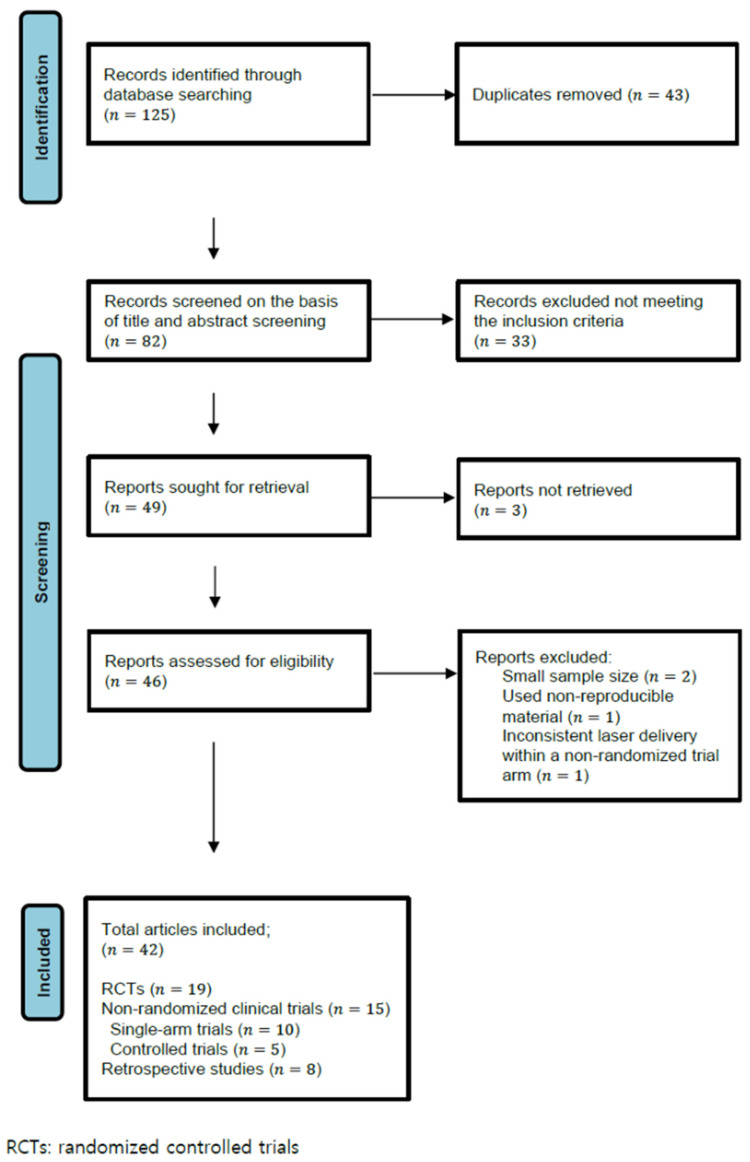
Literature search and article selection.

**Table 1 medicina-58-00936-t001:** Q-switched Nd:YAG laser with or without other treatment modality.

Year	Refs.	Study Design	Patients: nr, Ethnicity	Treatment A	Treatment B	Treatment C	Treat. Duration ^§^	Follow-Up Period ^†^	Melasma Type ^ǂ^	Efficacy/Outcomes *	Tolerability/ Adverse Events **	Recurrence Rates
2022	Micek I. et al. [12]	Prospective	40, Caucasian (Fitzpatrick II–III)	1064 nm QSNY (5 ns, 6–8 mm, 1.7–3.5 J/cm^2^, 2 passes for whole face, 4–8 passes at the discoloration site)	N/A	N/A	1–2 w, 9 s	B, +2 w (40/40) +12 M (21/40)	N/A	Significant decrease in mMASI (5.27 ± 2.47 to 3.54 ± 2.18), MI (183.17 ± 29.39 to 152.76 ± 24.91), and EI scores (351.96 ± 52.54 to 309.14 ± 46.01) 2 weeks after treatment Improvement of mMASI and MI scores maintained after 1 year	Temporary darkening of the hyperpigmentation (5/40), permanent discoloration (1/40), dryness (4/40)	8/21 (38%) 1 year after last session, during summertime
2022	Hong J.K. et al. [13]	Prospective, split-face	20, Korean (Fitzpatrick III–IV)	1064 nm QSNY (8 mm, 2.0–3.0 J/cm^2^, 10 Hz, 3–5 passes)	1064 nm PSNY (10 mm, 1.5–2.5 J/cm^2^, 10 Hz, 3–5 passes)	N/A	2 w, 5 s	B, +4 w	N/A	Significant decrease in mMASI score (both A and B) 4 weeks after treatment Clinical improvement (both A and B) 4 weeks after treatment No significant differences in mMASI and MI scores 4 weeks after treatment between A and B Patient satisfaction: in A, 15/19 (78.9%) had grade 4 (marked improvement) or 5 (very marked), in B 13/19 (67.4%) showed same score	No serious adverse events	N/A
2021	Ibrahim, S.M.A. et al. [14]	Prospective, randomized	50 Egyptian	1064 nm QSNY (8 mm, 1–1.5 J/cm^2^, 10 Hz, 2–6 passes)	Topical silymarin cream 1.4% (14 mg/mL): stearic acid 15 g, glycerin 5 g, KOH 0.72 g, H2O 79 g	N/A	(A) 2 w, 6 s (B) BID, 3 M	B, +3 M	A: D (3), E (4), M (18) B: D(4), E(4), M(17)	Significant decrease in mMASI score (both A and B) 12 weeks after treatment (A: 7.3 to 2.1, B: 9 to 2.3) No significant differences in mean difference and percentage of change of mMASI score 12 weeks after treatment between A and B (68.8%, 54.9%)	Worsening of melasma (1 in group B, but the patient did not use sun block properly)	N/A
2021	Esmat, S. et al. [15]	Prospective, split-face randomized	30, Egyptian	1064 nm QSNY (3 J/cm^2^, 6 mm, 10 Hz, 2 passes)	Group A: low power fractional CO_2_ alone (10 W, 800 micrometer interdot space, dwell time of 200 microseconds, no stacking) Group B: combined QSNY toning with low power fractional CO_2_	N/A	QSNY: 2 w, 9 s Low power fractional CO_2_: 4 w, 3 s	B, +1 w, +2 M, +3 M	N/A	Significant decrease in mMASI score 8 weeks after treatment with all regimens (QSNY 2.38 ± 1.49 to 1.49 ± 1.30, fractional CO2 2.04 ± 1.14 to 1.50 ± 0.86, combined 2.16 ± 1.43 to 1.04 ± 1.19), but did not maintain the reduction until 12 weeks after treatment Significant reduction in the MI scores 8 weeks after treatment with all regimens (QSNY 619.07 ± 45.60 to 613.83 ± 37.10, fractional CO_2_ 615.13 ± 47.37 to 636.29 ± 40.31, combined 621.80 ± 49.97 to 615.38 ± 30.78) Significantly greater reduction in mMASI and MI scores on the side receiving QSNY than low-power fractional CO_2_ after 1 week of treatment (64.03%, 8.27% vs. 36.02%, 2.64%), but the difference between becoming insignificant 8 weeks after treatment No significant differences in changes of mMASI and MI scores 8 weeks after treatment between QSNY and combined modality	Vitiligo-like depigmentation on the QSNY side (familial history of vitiligo) (1) MH only in the QSNL toning side (4) (1 in group A and 3 in group B)	N/A
2021	Debasmita, B. et al. [16]	Prospective, randomized	60, Indian	1064 nm QSNY (0.8 J/cm^2^, 4 Hz, 2.5 and 4 mm spot, 2 passes each) and topical 3% tranexamic acid	Microneedling (1.5 mm depth) and topical 3% tranexamic acid	N/A	4 w, 5 s	B, +2 M	N/A	Significant reduction in mMASI score after last treatment (both A and B) (A: 5.12 ± 2.66 to 2.33 ± 1.33, B: 4.60 ± 2.38 to 1.88 ± 1.08) No significant differences in changes of mMASI and MI scores after last treatment between A and B No significant differences in patient satisfaction after last treatment between A and B	Transient burning sensation: A (6/30, 20%), B (4/30, 13.3%) Transient pain: A (4/30, 13.3%), B (8/30, 26%) Erythema: A (2/30, 6.6%), B (6/30, 20%)	N/A
2021	Agamia N. et al. [17]	Prospective	60, Egyptian	Oral TA (250 mg/day)	1064 nm QSNY (4 mm, 2 J/cm^2^, 3 Hz) and oral TA (250 mg/day)	N/A	QSNY: 2 w, 6 s TA: 3 M	B, +0, +3 M	A: E (4), D (2), M (24) B: E (6), D (2), M (22)	Significant decrease in mMASI score (both A and B) after last treatment and at the end of follow-up (3 months) Group B showed a statistically higher change of mMASI score compared to group A after last treatment (in A, 23 ± 11.9, 16.7 ± 0, 20.9 ± 8.7 with epidermal, dermal, and mixed type, respectively, in B, 37.2 ± 12, 4.2 ± 0, and 41.1 ± 21.7, respectively)	Minimal, transient adverse events (unspecified)	N/A
2020	Dev, T. et al. [18]	Prospective, split-face, randomized	28, Indian (Fitzpatrick IV, V)	1064 nm QSNY (6 mm, 1.5 J/cm^2^, 10 Hz, 10 passes)	TC cream (hydroquinon 4%, tretinoin 0.05%, and fluocinolone acetonide z0.01%)	N/A	A: 1 w, 12 s B: QD, 12 w * Stopped if near-resolution was reached	B, +0, +1 M, +2 M, +3 M	N/A	Significant decrease in mMASI score (both A and B) 12 weeks after treatment (A: 3.3 ± 1.9 to 2.7 ± 1.5, B: 3.3 ± 2.0 to 2.3 ± 1.6) Significant decrease in MI score (both A and B) 12 weeks after treatment (A: 50.6 ± 5.9 to 48.3 ± 5.9, B: 49.9 ± 6.1 to 47.8 ± 5.4) Significant improvement in photographic assessment 12 weeks after treatment (both A and B) (A: 17.3%, B: 20.9%) Significant decrease in patient-reported severity score 12 weeks after treatment (both A and B) (A: 5 to 3.5 ± 0.9, B: 5 to 3.3 ± 1.1) No inter-modality difference in any of the above outcome measures	A: None, B: Erythema, and telangiectasia	All cases recurred in 21 patients of both groups who were followed up for 3 months
2019	Kwon, H.H. et al. [19]	Retrospective	114, Korean (Fitzpatrick III–V)	1064 nm QSNY (7–8 mm, 3.0 J/cm^2^ with PTP mode, 10 Hz) +FMR (0.5–1.00 mm depth, 20–30 intensity, 30–50 ms, 1–2 passes)	1064 nm QSNY (7–8 mm, 3.0 J/cm^2^ with PTP mode, 10 Hz)	N/A	1 w, 10 s	B, +3 M	N/A	Significantly greater reduction in mMASI score after last treatment in group A compared to group B (A: 2.9, B: 1.8) Better PhGA scoring after last treatment in group A compared to group B; A (excellent 29%, good 39%), B (excellent 14%, good 40%) Significantly higher patients’ self-assessment scoring after last treatment in group A than group B (A: 3.3, B: 2.2)	Significantly higher rates of MH in group B (8/58) compared to group A (5/56) Significantly higher rates of RH in group B (9/58) compared to group A (5/56) Significantly higher pain sensation in group A compared to group B (mean VAS, 3.8 vs. 2.1)	N/A
2019	Jung J.W. et al. [20]	Prospective, split-face	15, Korean	1064 nm QSNY (8 mm, 1.19 J/cm^2^ with PTP mode, median shot 2350.1) +FMR (50% intensity, 1 mm depth, 50 ms, 1 pass)	1064 nm QSNY (8 mm, 1.19 J/cm^2^ with PTP mode, median shot 2350.1)	N/A	2 w, 5 s	B, +2 w	N/A	Significant reduction in Mexameter^®^ (MI + EI) and PSI scores after last treatment (both A and B) Significantly greater reduction in Mexameter^®^ score after last treatment in group A compared to group B (A: 148.98 ± 57.45 to 65.55 ± 26.24, B: 143.24 ± 54.85 to 87.31 ± 42.13) Significantly greater reduction in PSI score after last treatment in group A compared to group B (A: 10.20 ± 3.06 to 5.50 ± 2.05, B: 11.13 ± 4.88 to 7.63 ± 3.16) Better patient satisfaction scoring after last treatment in group A compared to group B (A: 3.53 ± 0.81, B: 3.13 ± 0.96)	None of severe adverse events Significantly higher pain sensation in group A compared to group B (mean VAS, 3.4 vs. 1.73)	N/A
2018	Choi, J.E. et al. [21]	Retrospective	40, Korean (Fitzpatrick III–IV)	1064 nm QSNY (8 mm, 1.2–2.0 J/cm^2^, 10 Hz, more than 5 passes)	N/A	N/A	1 w, 10 s	B, +0, +3.6 ± 1.1 w	N/A	Significant decrease in mMASI score (3.19 ± 2.64 to 1.46 ± 1.06, 54.2%) after last treatment Improved in PhGA after last treatment; excellent (2.5%), good (35%), fair (37.5%), poor (15%) and no improvement (10%) The reduction in mMASI score after last treatment significantly increased relative to the number of treatment sessions	MH and RH (2/40, 5%)	N/A
2018	Kong, S.H. et al. [22]	Prospective, randomized, split-face	17, Korean (Fitzpatrick III–V)	1064 nm QSNY (7 mm, 1.2–2.0 J/cm^2^,10 Hz, 5–7 passes)	PDL + QSNY (firstly QSNY on the entire face and subsequent PDL 595 nm, 20 ms, 7 mm, 7–8 J/cm^2^, 2–3 passes) on the half of the face	N/A	A: 1 w, 9 s B: 1 w, 9 s (QSNY) + 4 w, 3 s (PDL)	B, +1 w, +2 M	N/A	Significant decrease in MASI score (both A and B) 8 weeks after treatment (A: 6.53 ± 2.65 to 5.07 ± 2.58, B: 6.23 ± 2.81 to 5.33 ± 2.84) No significant difference in MASI score between A and B 8 weeks after treatment Significant difference in MASI score between patients (n = 7) who showed visibly widened capillaries in dermoscopy between A and B 8 weeks after last treatment (A: 5.99 ± 2.86 to 5.09 ± 2.88, B: 6.60 ± 2.66 to 4.29 ± 2.24) No significant difference in MASI score in the patients (n = 10) who did not show visibly widened capillaries between A and B 8 weeks after last treatment	PIH, RH (2/17) only in group B (Fitzpatrick IV–V, who had visibly widened vessels in dermoscopy)	N/A
2018	Saleh, F. et al. [23]	Prospective, split-face	19, Egyptian (Fitzpatrick III–IV)	1064 nm QSNY (6–7 mm, 1.2–3.5 J/cm^2^, 10 Hz, 2–5 passes)	QSNY + modified Jessner’s solution peel (17% lactic acid, 17% salicylic acid, 8% citric acid dissolved in 95% ethanol)	N/A	(A) 2 w, 6 s (B) QSNY 3 s + peel 3 s (performed in alternating sequence, 2 w)	B, +1 M	M(19)	Significant decrease in mMASI score (both A and B) 1 month after last treatment (A: 46.3%, B: 47.5%) No significant difference in mMASI score between A and B 1 month after last treatment Significant decrease in amount of melanin pigment (presented in MPSA) (both A and B) after last treatment, with no significant difference between A and B Significant decrease in the number of MART-1-positive melanocytes after last treatment (both A and B), with no significant difference between A and B	MH (4/19, in the side A)	N/A
2017	Kaminaka C. et al. [24]	Randomized, split-face	13, Japanese (Fitzpatrick III–IV)	1064 nm QSNY (6 mm, 2.0–2.5 J/cm^2^, 5 Hz, 3 passes)	No treatment	N/A	1 w, 10 s	B, +0, +1 M, +3 M, +6 M	N/A	Significant decrease in mean MI score at 1 month after last treatment (214.5 ± 8.6 to 168.3 ± 8.2) Improvement of mean MI score maintained significantly until 6 months after last treatment Significant decrease in EI score 6 months after last treatment (291.1 ± 15.2 to 244 ± 17.0) Poor cases of melasma had higher EI score compared to the Good or Better cases at baseline (340.5 ± 14.7 vs. 264.66 ± 20.1)	PIH (1/20, 5.0%): spontaneously resolved after 3 months	1/12 (8.3%) in 3 months follow-up and 2/12 (16.7%) in 6 months follow-up
2017	Alavi, S. et al. [25]	Prospective, randomized	41, Iranian	1064 nm QSNY (400–500 mJ, 8 mm, 0.769–0.995 J/cm^2^)	QSNY + FEYL (400 mJ, 7 mm, 1.040 J/cm^2^, 10 Hz)	N/A	2 w, 4 s	B, +0	N/A	Significant increase in percent changes in Visioface^®^ score (both A and B) after last treatment (A: 29.25 ± 13.20%, B: 56.95 ± 40.29%) Significantly higher increase in percent change in Visioface^®^ score in group B after last treatment compared to group A Significantly higher decrease in percent change of MI score in group B after last treatment compared to group A (22.01 ± 10.67 vs. 7.69 ± 4.75) Significant decrease in EI score only in group B after last treatment (349 ± 62.53 to 320.47 ± 43.72) with no significant decrease in EI in group A	None of severe adverse events	N/A
2017	Jang, H.W. et al. [26].	Prospective, randomized, split-face	28, Korean (Fitzpatrick III–V)	Dual-pulsed 1064 nm QSNY (8 ns, 7 mm, 1.4 J/cm^2^, irradiated at dual pulses of 0.7 J/cm^2^, 80 μs intervals, 1000 shots)	Single-pulsed 1064 nm QSNY (6 ns, 7 mm, 1.4 J/cm^2^, 1000 shots)	N/A	1 w, 8 s	B, +0	N/A	Significant decrease in mMASI score (both A and B) after last treatment (A: 5.88 ± 3.16 to 2.52 ± 2.52, B: 5.89 ± 3.15 to 2.45 ± 2.25) Significant increase in L* values (both A and B) after last treatment (A: 62.22 ± 2.73 to 1.30 ± 1.62, B: 62.15 ± 2.99 to 1.17 ± 1.59) No patient-reported differences in patients’ satisfaction between A and B Direct comparisons between A and B after last treatment in objective outcome measures were absent	Significantly higher pain sensation in group B compared to group A (mean VAS, 3.3 vs. 4.6) None of pigmentary adverse events such as RH and MH	N/A
2016	Gokalp, H et al. [27]	Retrospective	34, Turkish (Fitzpatrick II–IV)	1064 nm QSNY (6 mm, 2.5 J/cm^2^)	N/A	N/A	2 w, 6–10 s	B, +0, +12 M	N/A	Significant decrease in mMASI score after last treatment (6.7 ± 3.3 to 3.2 ± 1.6) Patient satisfaction: in A, 20/34 (58.8%) rated themselves having at least a 50% reduction in melasma severity after last treatment	None of severe adverse events	20/34 (58.8%), 1 year after last session
2016	Hofbauer Parra, C.A. et al. [28]	Prospective	20, Brazilian (Fitzpatrick III–V)	1064 nm QSNY (8 mm, 0.8–1.6 J/cm^2^, 10 Hz, 1–3 passes to mild erythema)	N/A	N/A	1 w, 10 s	B, +1 w, +1 M, +3 M, +6 M	N/A	Significant decrease in mMASI score 1 week and 1 month after last treatment compared with the baseline (7.85 ± 4.24 to 4.33 ± 2.89, 6.43 ± 4.48) No significant decrease in mMASI score 3 and 6 months after last treatment (7.85 ± 4.24 to 7.92 ± 4.49, 7.49 ± 4.51) Histopathologically, a slight, nonsignificant decrease in melanin deposition seen in all layers of the epidermis 1 week after last treatment	N/A	13/16 (81%), 3 months after last session
2015	Vachiramo n, V. [29]	Prosepictve, randomized, split-face	15, Thai (Fitzpatrick III–V), all male	1064 nm QSNY (6 mm, 2.2–2.8 J/cm^2^, 10 Hz)	30% GA peeling + QSNY	N/A	1 w, 5 s	B, +1 M, +2 M, +3 M	N/A	Significantly lower RL*I in group B compared with A after last session (A: 7.98 ± 0.73 to 6.42 ± 0.63, B: 8.20 ± 0.73 to 4.35 ± 0.63) Significant decrease in mMASI score in group B (20.08 ± 1.99 to 13.00 ± 2.17) after last session with no significant decrease in group A Percentage of patients who rated their response as >75% clearing of melasma: at 4-week follow-up (A: 15.4%, B: 61.5%), at 12-week follow-up (A: 0%, B: 41.7%)	PIH (1), MH (1)	N/A
2015	Choi, C. P. et al. [30]	Retrospective	360, Korean (Fitzpatrick III–V)	1064 nm QSNY (6 mm, 2.5–3.0 J/cm^2^, 10 Hz)	1064 nm QSNY (6 mm, 2.1–2.5 J/cm^2^, 10 Hz) + LPNY (7 mm, 0.3 ms, 15–17 J/cm^2^, 5 Hz)	N/A	1 w, 10 s	B, +2 M	N/A	Significantly superior improvement in mMASI score 2 months after last session in group B (median 3.6) compared to group A (median 3.0) Significantly superior improvement in PhGA in group B compared to group A 2 months after last session	MH, RH (A: 25/177, 14.1%, B: 2/183, 1.1%)	N/A
2015	Choi, C. P. et al. [31]	Retrospective	30, Korean (Fitzpatrick III–IV), who have aggravated after previous dual toning treatment)	1064 nm QSNY (6 mm, 2.1–2.5 J/cm^2^, 10 Hz) + LPNY (0.3 ms, 7 mm, 15–17 J/cm^2^, 5 Hz)	N/A	N/A	1 w, 10 s, then maintenance (2 w, 4 s, 4 w, 3 s, 12 w, 1 s)	B, +2 M (before maintenance)	N/A	Significant decrease in mMASI score (10.48 ± 3.64 to 3.22 ± 1.45) 2 months after last session PhGA: 76–100% improvement (80%), 51–75% improvement (20%)	None of pigmentary adverse events such as RH and MH	N/A
2014	Yun, W.J. et al. [32]	Prospective, randomized	24, Korean (Fitzpatrick III–IV)	IPL (560–800 nm, 13–15 J/cm^2^)	IPL (560–800 nm, 13–15 J/cm^2^) + QSNY (5–10 ns, 6 mm, 1.5–2.0 J/cm^2^, 10 Hz, 4–6 passes)	N/A	2 w, 6 s	B, +1 M, +2 M	N/A	Significantly greater decrease in partial MASI (for cheeks) score 2 months after treatment in group B compared to group A (A: 12.0 ± 3.33 to 9.17 ± 2.86, B: 12.75 ± 4.58 to 6.50 ± 3.29) Significant decrease in percent change of partial MASI score after treatment in group B compared to the baseline (47% at 1 month & 50% at 2 months after treatment) Significant decrease in MI score in group B (20.1%) with no significant decrease in group A 2 months after treatment Insignificant reduction in EI score (Both A and B) 2 months after treatment	1st degree burn (1 in group B)	N/A
2014	Alsaad, S.M. et al. [33]	Prospective, randomized, split-face	10, Ethics unspecified (Fitzpatrick II–V)	Microdermabration +1064 nm QSNY (50 ns, 5–6 mm, 1.6 J/cm^2^, 4 Hz, 2 passes) +0.05% fluocinolone cream	Microdermabration +1064 nm QSNY (5 ns, 5–6 mm, 1.6 J/cm^2^, 4 Hz, 2 passes) +0.05% fluocinolone cream	N/A	4 w, 3 s	B, +1 M, +3 M, +6 M	N/A	Significant decrease in MASI score (both A and B) 1 month after last session (A: 35%, B: 28%) Significant decrease in MASI score (both A and B) 6 months after last session (A: 28%, B: 23%) No significant difference in MASI score between A and B 1 and 6 months after last session	Significantly higher pain sensation in the group B compared to group A (mean NRS, 1.2 vs. 2.9)	At 3 months after last session, reduction in MASI score was insignificant from baseline in both group A and group B (A: 12%, B: 11%)
2014	Fabi S.G. et al. [34]	Prospective, randomized, split-face	20, Ethics unspecified (Fitzpatrick II–IV)	1064 nm QSNY (8 mm, 1–2 J/cm^2^, 5 Hz, 1–8 passes)	755 nm QSAL (6–8 mm, 1.8 J/cm^2^, 5 Hz, 1–2 passes)	N/A	1 w, 6 s	B, +2 w, +3 M, +6 M	M(20)	Significant improvement in mMASI score (both A and B) 24 weeks after last session No significant difference of mMASI score between A and B at any visit until 24 weeks after last session No significant difference of patients’ self-assessment between A and B	No serious adverse events	N/A
2014	Sim, J.H. et al. [35]	Prospective	50, Korean	1064 nm QSNY (8 mm, 2.8 J/cm^2^, 10 Hz)	N/A	N/A	1 w, 15 s	B, +0	N/A	Significant improvement in pigmentation levels by Janus pigment imaging technology system found after last session ([19.66, 18.70, 17.60] to [15.62, 14.12, 13.32] on front, left, and right side, respectively) Both patients and investigators rated treatment outcome as “good improvement” on average with improvement rate of 50–74%	No serious adverse events	N/A
2014	Lee, D.B. et al. [36]	Prospective, randomized	52, Korean	1064 nm QSNY (7 mm, 1.0–1.7 J/cm^2^, 10 Hz)	1064 nm QSNY + Jessner’s peel (salicylic acid 14 g, resorcinol 14 g, lactic acid 14 g dissolved in 95% ethanol)	N/A	2 w, 10 s	B, +0	N/A	Significant decrease in MASI score (Both A and B) after last session (A: 8.68 ± 4.06 to 6.22 ± 2.54, B: 8.98 ± 3.72 to 6.05 ± 2.66) No significant difference in reduction in MASI, patients’ self-assessment and PhGA between A and B after last session	Burning sensation in Group B (4/26)	N/A
2013	Shin, J.U. et al. [37]	Prospective, randomized	48, Korean (Fitzpatrick III–IV)	1064 nm QSNY (7 mm, 2 J/cm^2^)	QSNY 1064 nm (2 J/cm^2^, 7 mm) + oral TA (750 mg/day)	N/A	A: 4 w, 2 s B: 4 w, 2 s + concurrently oral TA 8 w	B, +1 M	N/A	Significant decrease in mMASI score (both A and B) 1 month after last session (A: 7.9 ± 3.9 to 6.0 ± 3.2, B: 7.9 ± 3.7 to 5.0 ± 3.4) Significantly greater reduction in percent change of mMASI score 1 month after last session in group B compared to group A (A: 21.9 ± 18.5%, B: 37.8 ± 23.9%) PhGA: 2/24 (9%) in group A and 5/24 (22%) in group B reported ≥50% improvement 1 month after last session	Oral TA associated gastrointestinal adverse events: heartburn (2, 4.2%), nausea (1, 2.1%)	N/A
2013	Na, S.Y. et al. [38]	Retrospective	35, Korean (Fitzpatrick III–IV)	IPL (10–10.5 J/cm^2^, 2.5 ms, delay time 10 ms between pulses, double pulses, 555–950 nm) after two weeks, 1064 nm QSNY (2.0–2.5 J/cm^2^, 6 mm, 10 Hz, 7–8 passes)	1064 nm QSNY (2.0–2.5 J/cm^2^, 6 mm, 10 Hz, 7–8 passes)	N/A	A: IPL 1 time, followed by QSNY 1 w, 4 s (2 w between IPL and QSNY) B: 1 w, 5 s	B, +1 w	M(35)	Significant decrease in MI, EI, and mMASI scores (both A and B) 1 week after last session; in MI, A: 174.08 ± 64.32 to 128.65 ± 41.36, B: 148.80 ± 35.29 to 130.33 ± 28.63, in EI, A: 295.05 ± 47.34 to 238.40 ± 48.67, B: 287.60 ± 55.87 to 255.80 ± 55.87, in mMASI, A: 8.54 to 3.52, B: 7.48 to 3.99 Significantly greater decrease in percent change of MI score 1 week after last session in group A compared to group B (A: 45.44 ± 35.71%, B: 18.47 ± 20.73%) Significantly greater decrease in percent change of mMASI score 1 week after last session in group A compared to group B (A: 59.35 ± 14.94%, B: 45.66 ± 14.75%)	None of pigmentary adverse events such as RH and MH	No recurrence at mean 5.9 months after last session in 12/20 of group A
2013	Kim, H.S. et al. [39]	Prospective, randomized, split-face	26, Korean (Fitzpatrick III–IV)	1064 nm QSNY (1.2–1.4 J/cm^2^, 6 mm, 10 Hz)	1064 nm QSNY (1.2–1.4 J/cm^2^, 6 mm, 10 Hz) +1550 nm FEGL (dynamic mode, 6–8 mJ/MTZ, MTZ diameter of 150 um, total density 300 mTZs/cm^2^)	N/A	QSNY: 2 w, 10 s FEGL: 4 w, 5 s	B, +1 M, +3 M	N/A	Significant decrease in mMASI score 4 and 12 weeks after last session in group B from baseline (4.40 ± 1.57 to 1.47 ± 0.66, 1.85 ± 0.83) Significant decrease in mMASI score 4 and 12 weeks after last session in group A from baseline (4.35 ± 1.41 to 1.51 ± 0.61, 1.77 ± 0.78) No significant difference in mMASI and PhGA between A and B 4 and 12 weeks after treatment Patients’ self-assessment: 65.4% of group A and 73.1% of group B rated themselves as definitely improved	Transient PIH (2, Fitzpatrick V)	N/A
2012	Na, S.Y. et al. [40]	Retrospective	20, Korean (Fitzpatrick III–IV)	IPL (10–10.5 J/cm^2^, 2.5 ms, delay time 10 ms between pulses, double pulses) after two weeks, 1064 nm QSNY (2.0–2.5 J/cm^2^, 6 mm, 10 Hz, 7–8 passes)	N/A	N/A	IPL 1 time, followed by QSNY 1 w, 4 s (2 w between IPL and QSNY)	B, +1 w	M(20)	Significant decrease in MI and EI scores after last session (174.08 ± 64.32 to 128.65 ± 41.36, 295.05 ± 47.34 to 238.40 ± 48.67) Significant decrease in MASI score after last session (8.54 to 3.52, 59.4%)	None of severe adverse events	N/A
2012	Kauvar, A.N.B. [41]	Prospective	27, Ethics unspecified (Fitzpatrick II-V), refractory to previous treatment (topical, chemical peel, laser)	Microdermabrasion(2 passes over entire face) followed by 1064 nm QSNY (5–7 ns, 1.8–2 J/cm^2^, 6 mm, in 10 patients, 50 ns, 1.6 J/cm^2^, 5 mm, in 17 patients) Skin care of hydroquinone 4% BID, 0.05% tretinoin QD or 15% L-ascorbic acid QD	N/A	N/A	4 w, 6 s	B, +3 M, +6 M, +12 M	M(27)	PhGA: mean clearance scores (at 3 months follow-up, 3.3, at 6 months follow-up, 3.2, and at 12 months follow-up, 3.3) The correlation between skin type and the percent clearance not significant * Clearance score: 3 = 76–95% improvement, 4 = >95% improvement	None of pigmentary adverse events such as RH and MH Mild irritation from skin care regimen (4/27, 15%)	N/A
2012	Bansal, C. et al. [42]	Prospective, randomized	60, Indian (Fitzpatrick III–V)	1064 nm QSNY (0.5–1 J/cm^2^, 6–8 mm, 10 Hz, 10 passes, fluence increased by 0.1 J/week until 1 J/cm^2^)	20% Azelaic acid (AA) cream	Combination of A and B * AA cream not applied on the day of the laser therapy	QSNY: 1 w, 12 s AA: BID, 3 M	B, +0	A: E(3), D(4), M(13) B: E(2), D(6), M(12) C: E(3), D(2), M(15)	Significant decrease in MASI score in all treatment regimens (A, B, C) after last session (A: 21.11 ± 6.91 to 10.11 ± 4.28, B: 15.90 ± 5.49 to 9.68 ± 3.37, C: 18.73 ± 7.53 to 4.94 ± 1.67) Significantly greater improvement of MASI score after last session in group C compared to group A and group B No significant difference of reduction in MASI score after last session between group A and group B	Burning sensation (2, 1 in B, 1 in C), erythema (1, in C)	N/A
2011	Zhou, X. et al. [43]	Prospective	50, Chinese (Fitzpatrick III–IV)	1064 nm QSNY (2.5–3.4 J/cm^2^, 6 mm, 10 Hz, 5 passes)	N/A	N/A	1 w, 9 s	B, +3 M	E(35), D(6), M(9)	Significant decrease in MI, MASI scores after last session (69.9 to 44.9, 10.6 ± 5.6 to 4.1 ± 3.9) Significant decrease in percent change of MI, MASI scores after last session (35.8%, 61.3%) Patients’ self-assessment: excellent (54%), good (30%), fair (60%), poor (10%)	None of severe adverse events	32/50 (64%), in 3 months follow-up
2011	Suh, K.S et al. [44]	Prospective	23, Korean (Fitzpatrick III–V)	1064 nm QSNY (5–7 ns, 3–4 J/cm^2^ for Fitzpatrick III-IV, 2–3 J/cm^2^ for Fitzpatrick V, 4/6/8 mm, 10 Hz)	N/A	N/A	1 w, 10 s	B, +0, +1 M, +2 M, +3 M	E(4), M(19)	Significant decrease in MASI score after last session (14.15 ± 1.47 to 7.57 ± 2.91) and 1, 2, 3 months after last session (8.22 ± 2.90, 8.95 ± 2.92, 10.15 ± 2.70) Significant increase in L* 10 weeks after last session (60.71 ± 2.99 to 61.95 ± 2.14) and 1, 2, 3 months after last session (61.73 ± 2.14, 61.59 ± 2.14, 61.26 ± 2.52) Significant increase in patient’s satisfaction score after last session (2.11 ± 1.01 to 8.88 ± 1.18) and 1, 2, 3 months after last session (7.53 ± 1.40, 7.38 ± 1.41, 7.02 ± 1.34)	Prolonged erythema (3/23) PIH (3/23), MH (1/23)	N/A
2011	Park, K.Y. et al. [45]	Prospective, randomized, split-face	16, Korean	1064 nm QSNY (2.0–2.3 J/cm^2^, 6 mm, 10 Hz)	1064 nm QSNY (2.0–2.3 J/cm^2^, 6 mm, 10 Hz) + 30% GA peel (1–2 min)	N/A	QSNY: 1 w, 6 s Peel: 2 w, 3 s	B, +0, +1 M, +2 M, +3 M, +4 M, +5 M	N/A	Significant decrease in MI and mMASI scores after last session (both A and B); in MI, A: 198.06 ± 31.56 to 162.40 ± 24.26, B: 198.41 ± 33.92 to 149.69 ± 30.11, in mMASI, A: 20.7 ± 1.8 to 17.2 ± 1.9, B: 21.2 ± 1.7 to 15.4 ± 1.5) Significantly greater improvement in percent change of MI after last session in group B compared to group A (32.6% vs. 22.0%) PhGA: >50% improvement (A: 31%, B: 69%) 5 months after last session Patients’ self-assessment: 38% of group A and 75% of group B rated themselves in good or excellent improvement 5 months after last session	None of severe adverse events	N/A
2011	Kar, H.K. et al. [46]	Prospective, randomized	75, Indian	1064 nm QSNY (0.5–1 J/cm^2^, 6–8 mm, 10 Hz, 10 passes, fluence increased by 0.1 J/week until 1 J/cm^2^)	35% GA peel 1/2/3 min for first 3 sessions, 70% GA peel 1/2/3 min for remaining 3 sessions	Epidermal type: 532 nm QSNY (0.5–1 J/cm2, 4 mm, 2 Hz) Mixed type: 1064 nm QSNY (2.0–2.5 J/cm^2^, 6 mm, 2 Hz, performed in the same session with 532 QSNY)	A: 1 w, 12 s B: 2 w, 6 s C: 2 w, 6 s	B, +0, +3 M	A: E(13), M(8) B: E(9), M(10) C: E(9), M(11)	Significant improvement in MASI score immediately after treatment for all regimens (A: 13.54 ± 7.19 to 7.05 ± 5.24, B: 10.78 ± 6.05 to 6.43 ± 5.0, C: 10.57 ± 5.13 to 8.37 ± 4.18) Significantly greater improvement of MASI score immediately after treatment in group A compared to group C, and in group A compared to B, and in group B compared to group C Worsening of percent change of MASI score 12 weeks after last session in all regimens (13.04%, 13.13%, 13.25%)	MH (6) (1/21 in A, 5/20 in C) PIH (7) (1/19 in B, 6/20 in C)	N/A
2011	Kang, H.Y. et al. [47]	Prospective	30, Korean (Fitzpatrick IV)	$1064\;\mathrm{nm}\;\mathrm{QSNY}\;(5\;\mathrm{ns},\;1.2\;J/{\mathrm{cm}}^{2},\;8\;\mathrm{mm},\;10\;\mathrm{Hz},\;2–3\;\mathrm{passes},\;\mathrm{then}\;\mathrm{immediately}\;\mathrm{LPQY}\;(7.0\;J/{\mathrm{cm}}^{2},\;5\;\mathrm{mm},\;300\;\mu$s, 10 Hz, 2–3 passes)	N/A	N/A	2 w, 10–12 s	B, +0, +6 w	N/A	Patients’ self-assessment: 20/30 (67%) patients reported >25% improvement, 7/30 (23%) patients reported 11–25% improvement, 3/30 (10%) reported 0–10% improvement after last session and maintained until 6 weeks after last session	None of severe adverse events	N/A
2011	Brown, A.S. et al. [48]	Prospective	21, Ethics unspecified (Fitzpatrick II–IV)	1064 nm QSNY (3–4 J/cm^2^ for Fitzpatrick II, 2–3 J/cm^2^ for Fitzpatrick III–IV, 8–10 mm)	N/A	N/A	1 w, 8 s	B, +0, +3 M	E or M (numbers unspecified)	Significant decrease in MASI score 8 weeks after last session (4.43 to 1.51). The most significant improvement in MASI score seen between baseline and week 4 (38.6%) PhGA: 19/21 showed 25–100% improvement	N/A	Flare was common 3 months after last session
2010	Wattanakr ai, P. et al. [49]	Prospective, randomized, split-face	22, Thai (Fitzpatrick III)	Pretreated with 2% HQ cream QD for 2 weeks and followed by 1064 nm QSNY (3.0–3.8 J/cm^2^, 6 mm, 10 Hz)	2% HQ cream QD	N/A	QSNY: 1 w, 5 s HQ: QD	B, +0, +1 M, +2 M, +3 M	D or M (numbers unspecified)	Significant decrease in RL*I after last session in group A (4.6 ± 1.9 to 0.6 ± 1.3) Insignificant decrease in RL*I after 7 weeks of topical application in group B (4.3 ± 1.7 to 3.4 ± 1.6) Significantly greater reduction in improvement rate of RL*I after treatment in group A compared to group B Significant decrease in mMASI score after last session in group A (22.3 ± 1.8 to 5.7 ± 0.8) Insignificant decrease in mMASI score after 7 weeks of topical application in group B (21.9 ± 1.8 to 16.6 ± 1.4) Patients’ self-assessment: in group A, 86.4% rated >50% improvement, 13.6% rated 50–75% improved; in group B, 13.6% rated >50% improvement, 36.4% rated 50–75% improved, and 50% rated little or not improved	MH (3/22, Fitzpatrick V) RH (4/22 in 5 sessions, 8/22 in patients with additional 5–10 weekly QSNY after completing the study)	Partial recurrence (22/22) in 3 months follow-up
2010	Polnikorn, N. [50]	Prospective	35, Thai, refractory melasma	1064 nm QSNY (3.0–3.4 J/cm^2^, 6 mm, 10 Hz, 10 passes) + topical 7% alpha arbutin solution	N/A	N/A	QSNY: 1 w, 10 s, then, 4 w, 2 s Arbutin: BID	B, +2 w (before maintenance), +2 w (after maintenance)	D or M (numbers unspecified)	PhGA: 26–50% fading of melasma lesions (48.39%), 51–80% reduction in lesions (29.04%) at 10 weeks after treatment initiation PhGA: 51–80% reduction (36.67%), >81% reduction (30%) after additional two subsequent monthly treatments	MH (3/35, 8.6%, spontaneously resolved within a few months)	Recurrence (2/35, 5.7%)
2010	Jeong, S.Y. et al. [51]	Prospective, split-face, cross-over	13, Korean (Fitzpatrick III–IV)	Pretreated with TC cream (4% hydroquinone, 0.05% tretinoin, 0.01% fluocinolone acetonide) for 8 weeks and followed by 1064 nm QSNY (1.5–2.0 J/cm^2^, 7 mm, 2 passes)	1064 nm QSNY (1.5–2.0 J/cm^2^, 7 mm, 2 passes) and followed by TC cream for 8 weeks (reverse sequence of treatment A)	N/A	TC: QD QSNY: 1 w, 8 s	B, +1 w, +11 M	N/A	(Group A) 8 weeks of topical cream insignificantly reduced MASI score, and the following 8 weeks of QSNY significantly decreased MASI score (3.0 ± 4.14 to 2.09 ± 3.92) L* remained unchanged 8 weeks after topical treatment, but the following 8 weeks of QSNY significantly increased L* (58.74 ± 4.45 to 60.78 ± 4.44) ΔE*ab decreased insignificantly 8 weeks after topical treatment but decreased significantly with the following 8 weeks of QSNY (5.51 ± 2.92 to 3.86 ± 2.37) (Group B) 8 weeks of QSNY significantly decreased MASI score (3.20 ± 3.49 to 1.74 ± 3.93), the following 8 weeks of topical treatment rather increased MASI score (1.74 ± 3.93 to 2.22 ± 3.82) with insignificant overall improvement L* was insignificantly increased after 8 weeks of QSNY, and the following 8 weeks of topical treatment rather decreased L* insignificantly ΔE*ab decreased significantly 8 weeks after QSNY (4.96 ± 2.70 to 4.69± 2.45) but rather increased after the following 8 weeks of topical treatment	TC: RH (3/13), irritation (4/13)	In group B, 4/13 (30.8%) showed partial recurrence within 3 months after treatment After 11 months after treatment, mild aggravation (9/13, but no deterioration from the initial condition)
2010	Choi, M. et al. [52]	Prospective	20, Korean (Fitzpatrick III–IV)	1064 nm QSNY (2.0–3.5 J/cm^2^, 6 mm, 10 Hz)	N/A	N/A	1 w, 5 s	B, +1 M	N/A	Significant increase in L* score (59.57 ± 3.63 to 60.43 ± 3.03) Significant decrease in MI (201.69 ± 48.92 to 173.47 ± 33.48)	None of severe adverse events	N/A
2009	Cho, S.B. et al. [53]	Retrospective	25, Korean (Fitzpatrick IV)	1064 nm QSNY (2.5 J/cm^2^, 6 mm, 2 passes for entire face or both cheeks, 4–5 J/cm^2^, 4 mm, 2 passes for melasma lesions)	N/A	N/A	2 w, average 7 s (range 5–15 s)	B, +2 M	N/A	PhGA: 2/25 (8%) rated improvement <25%, 5/25 (20%) rated improvement of 25–50%, 11/25 (44%) rated improvement of 51–75%, 7/25 (28%) rated improvement of 76–100%Patients’ satisfaction: 18/25 (72%) rated very satisfied or satisfied, 5/25 (20%) rated slightly satisfied, 2/25 (8%) rated unsatisfied	MH (2/25, 8%, not accentuated on Wood’s light)	At least 3 out of 25, 2–6 months after last session

N/A: non-applicable, PhGA: Physician’s global assessment, MH: mottled hypopigmentation, RH: rebound hyperpigmentation, PIH: postinflammatory hyperpigmentation, PSI: pigmentation and severity index, MPSA: melanin particle substance area, RL*I: relative lightness index.; QSNY: Q-switched Nd:YAG laser, TC: triple combination, TA: tranexamic acid, PSNY: picosecond Nd:YAG laser, LPNY: long-pulsed Q-switched Nd:YAG laser, FEYL: fractional Er:YAG laser, FMR: fractional microneedling radiofrequency, IPL: intense pulsed light, QSAL: Q-switched alexandrite laser, FEGL: fractional Er:Glass laser, AA: azelaic acid, GA: glycolic acid, HQ: hydroquinone.; ^§^ w: week interval, s: session, M: months, QD: once daily, BID: twice daily; ^†^ B: baseline, +nw: n week(s) after the last session, +nm: n month(s) after the last session, +0 means that evaluation was performed immediately after the last session.; ^ǂ^ E: epidermal type, D: dermal type, M: mixed type. * In melasma area severity index (MASI), modified MASI (mMASI), melanin index (MI) and erythema index (EI) using Mexameter^®^, PSI (pigmentation and severity index), L*I (lightness index), RL*I (relative lightness index), Visioface^®^ score and ΔE*ab (color difference index), the lower the score, the milder the severity is. However, in L* (lightness) score and grade of improvement, the higher the score, the higher the severity is.; ** Transient erythema and swelling after QSNY were all excluded in this table because it usually resolves within minutes to hours spontaneously.

**Table 2 medicina-58-00936-t002:** Summaries of the commonly used outcome measures for evaluating melasma.

Outcome Measures	Definition	Calculation Methods
MASI	Melasma area and severity index	0.3 × A(forehead) × {D(forehead) + H(forehead)} + 0.3 × A(left malar) × {D(left malar) + H(left malar)} + 0.3 × A(right malar) × {D(right malar) + H(right malar)} + 0.1 × A(chin) × {D(chin) + H(chin)}A: area of involvement (0 = absent, 1 = <10%, 2 = 10–29%, 3 = 30–49%, 4 = 50–69%, 5 = 70–89%, and 6 = 90–100%) D: darkness (0 = absent, 1 = slight, 2 = mild, 3 = marked, and 4 = severe) H: homogeneity (0 = absent, 1 = slight, 2 = mild, 3 = marked, and 4 = severe)
mMASI	Modified melasma area and severity index	0.3 × A(forehead) × D(forehead) + 0.3 × A(left malar) × D(left malar) + 0.3 × A(right malar) × D(right malar) + 0.1 × A(chin) × D(chin) The abbreviations “A” and “D” are same as the above
MI	Melanin index	Values on an arbitrary unit (AU) (0–999) measured by reflectance spectrophotometer
EI	Erythema index	Values on an arbitrary unit (AU) (0–999) measured by reflection spectrophotometer
L*	The lightness	Values measured by colorimeter or spectrophotometer on a gray scale from 0 (black) to 100 (white)
L*I	Lightness index	Average of multiple L* measurements from different darkest areas measured by colorimeter or spectrophotometer
RL*I	Relative lightness index	The difference of the L*I between normal skin and melasma measured by colorimeter or spectrophotometer
ΔE*ab	Color difference	$\sqrt{{\left(\Delta L^{*}\right)}^{2}+{\left(\Delta a^{*}\right)}^{2}+{\left(\Delta b^{*}\right)}^{2}}$, which incorporates the difference of the L* (∆L*), a*(∆a*, difference in red and green), and b*(∆b*, difference in yellow and blue) between normal skin and melasma measured by colorimeter or spectrophotometer

**Table 3 medicina-58-00936-t003:** Summary of highlighted outcomes in this study.

▪ LT showed significant improvement in melasma with various rates of occasional MH, RH, or short-term recurrence as adverse events. ▪ LT showed superiority to GA peel, HQ cream, or AA cream in efficacy as a monotherapy. ▪ Although LT did not show superiority to TC or silymarin cream in efficacy, it seemed to be slightly safer. ▪ LT showed significantly better safety profile than 532 nm QSNY when treating melasma. ▪ LT using dual-pulsed QSNY (PTP mode) was as effective as conventional LT with better tolerability and safety profile. ▪ LT using PNSY did not show superiority to conventional LT in efficacy. However, data is still insufficient on this novel picosecond system. ▪ Combination therapy of LT and other EBDs such as FEYL, FMR, PDL, LPNY, and IPL showed superior efficacy to LT alone. In addition, combination with FMR or LPNY lowered the incidence of MH and RH. ▪ Combination therapy of LT and FCO_2_ or FEGL did not show superior efficacy over LT alone. However, combination with FCO_2_ lowered the incidence of MH. ▪ Combination therapy of LT and other non-EBDs such as GA peel, oral TXA, and AA cream showed superior efficacy to LT alone. ▪ Combination therapy of LT and Jessner’s or modified Jessner’s peel did not show superiority to LT alone in efficacy. However, combination with modified Jessner’s peel lowered the incidence of MH.

LT: laser toning, GA: glycolic acid, HQ: hydroquinone, AA: azelaic acid, TC: triple combination, QSNY: Q-switched Nd: YAG laser, MH: mottled hypopigmentation, RH: rebound hyperpigmentation, PTP: photoacoustic twin pulse, PSNY; picosecond Nd: YAG laser, EBD: energy-based device, FEYL: fractional Er: YAG laser, FEGL: fractional Er: Glass laser, FMR: fractional microneedling radiofrequency, LPNY: long-pulsed Nd: YAG laser, FCO2: fractional CO2 laser, TXA: tranexamic acid.

## Data Availability

Not applicable.

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
