# Peer review of "The Low-Fluence Q-Switched Nd:YAG Laser Treatment for Melasma: A Systematic Review"

_medicina, 2022, doi:10.3390/medicina58070936_

Round 1
Reviewer 1 Report
Please see the following comments:
This study is a systematic review of the low fluence QS Nd: YAG laser treatment for Melasma. Melasma is a very common acquired facial pigmentary disorder and it is very difficult to treat. In the most recent systemic review and meta-analysis of laser therapy in the treatment of melasma, published in Feb 2022 by Dihui Lai et al. (doi: 10.1007/s10103-022-03514-2. Epub 2022 Feb 5.), it is confirmed that the laser and laser-based combination treatment for melasma could significantly reduce the MASI score.
This systemic review is focused on the low fluence QS Nd: YAG laser treatment for Melasma. The study is well written and follows the PRISMA guidelines.
I recommend the authors adding more description of the technique in the introduction part for the readers who are not familiar with this method of the treatment. Moreover, authors can refer to the previous findings to the present.
The figure 1 belongs to the material and methods part, and it should be mentioned in this part.
In the material and methods part, the line 85, please add the number of the table.
In the results part, I recommend authors adding a table and provide more detail about the assessment scores used in the studies, such as, melasma area severity index (MASI), modified MASI (mMASI), melanin index (MI) and erythema index (EI), lightness index (L*I), relative lightness index (RL*I) and color difference index (ΔE*ab) using a spectrophotometer, and physician’s global assessment (PhGA).
In the dissection part, I recommend adding a table for the highlighted outcomes and the new findings, added to the previous studies.
Good luck.
Author Response
Thank you very much for the reviewers to review our manuscript and provide constructive feedback. We modified or responded to the reviewers' recommendations as follows.
Reviewer #1: This study is a systematic review of the low fluence QS Nd: YAG laser treatment for Melasma. Melasma is a very common acquired facial pigmentary disorder and it is very difficult to treat. In the most recent systemic review and meta-analysis of laser therapy in the treatment of melasma, published in Feb 2022 by Dihui Lai et al. (doi: 10.1007/s10103-022-03514-2. Epub 2022 Feb 5.), it is confirmed that the laser and laser-based combination treatment for melasma could significantly reduce the MASI score.
This systemic review is focused on the low fluence QS Nd: YAG laser treatment for Melasma. The study is well written and follows the PRISMA guidelines.
I recommend the authors adding more description of the technique in the introduction part for the readers who are not familiar with this method of the treatment. Moreover, authors can refer to the previous findings to the present
: Thank you for your invaluable comments. We agreed with your opinion and modified the introduction section as following: However, since 2000s, low-fluence Q-switched Nd:YAG laser (LFQSNY), commonly referred to as ‘laser toning (LT)’ has been accepted as a new gold standard of melasma treatment in Asia, where there is high demand for treatment. This technique involves multiple sessions (usually around 10 sessions) of weekly or biweekly 1064nm QSNY treatment with a low fluence (usually 1-3J/cm2), collimated beam with a large spot size and a frequency of 5-10 Hz. The endpoint of the procedure would be faint erythema. LT It is known to selectively destroy melanin in melanophores, while melanin-containing cells are left undamaged, resulting in safe depigmentation of melasma. [8,9]
The figure 1 belongs to the material and methods part, and it should be mentioned in this part.
: The figure 1 is fixed to be mentioned in the material and methods part.
In the material and methods part, the line 85, please add the number of the table.
: The table 1 is now noted in the abovementioned line.
In the results part, I recommend authors adding a table and provide more detail about the assessment scores used in the studies, such as, melasma area severity index (MASI), modified MASI (mMASI), melanin index (MI) and erythema index (EI), lightness index (L*I), relative lightness index (RL*I) and color difference index (ΔE*ab) using a spectrophotometer, and physician’s global assessment (PhGA).
: In accordance with your opinion, we added a table (table 2) to render information about the outcome measures commonly used for evaluating melasma.
In the dissection part, I recommend adding a table for the highlighted outcomes and the new findings, added to the previous studies. Good luck.
: Efficacy, tolerability, and adverse events of LT monotherapy and combination therapy with energy-based device (EBD) or non-EBD is briefly summarized in the box (table 3). We hope our new tables help readers fully understand our manuscript.
Thank you for your consideration of this manuscript.

Reviewer 2 Report
As I stated in the comments I believe this article covers an interesting topic that is useful for the clinicians. The authors could have included other search engines in the literature review, however, I believe the data you gathered is good and valuable.
Table no 1 is complex, with important features included such as phototype and adverse effects. I also appreciate the fact that authors included not only RCT; even though studies were heterogenous, with variable outcome measures, reported results may be valuable for the clinicians.
Other search engines could have also been used, but I believe the study is well-performed and valuable. Congratulations to the authors!
Author Response
Thank you very much for the reviewers to review our manuscript and provide constructive feedback. We modified or responded to the reviewers' recommendations as follows.
Reviewer #2: As I stated in the comments I believe this article covers an interesting topic that is useful for the clinicians. The authors could have included other search engines in the literature review, however, I believe the data you gathered is good and valuable.
Table no 1 is complex, with important features included such as phototype and adverse effects. I also appreciate the fact that authors included not only RCT; even though studies were heterogenous, with variable outcome measures, reported results may be valuable for the clinicians.
Other search engines could have also been used, but I believe the study is well-performed and valuable. Congratulations to the authors!
: Thank you for your careful review of our manuscript and for your generous encouragement and advices. We hope our review article help physicians get a better understanding about laser toning and make an optimal treatment plan for melasma.
Thank you for your consideration of this manuscript.

Reviewer 3 Report
An interesting systematic review about the use of low-fluence Q-switched Nd: YAG laser treatment for melasma. I have found a problem to assess before publication:
The study selection process must be corrected; there is an error in figure 1: you stated that you excluded 36 studies after abstract screening; so the number of studies should reduce from 82 to 46, and then after excluding the article not retrieved to 43 ; please check and correct figure.
Author Response
Thank you very much for the reviewers to review our manuscript and provide constructive feedback. We modified or responded to the reviewers' recommendations as follows.
Reviewer #3: An interesting systematic review about the use of low-fluence Q-switched Nd: YAG laser treatment for melasma. I have found a problem to assess before publication:
The study selection process must be corrected; there is an error in figure 1: you stated that you excluded 36 studies after abstract screening; so the number of studies should reduce from 82 to 46, and then after excluding the article not retrieved to 43 ; please check and correct figure.
: Thank you for your invaluable comments. We identified the errors and fixed them in both the text and the figure 1 as following:
A total of 125 articles were initially identified in the literature search, of which 43 were duplicates and 33 did not meet the inclusion criteria and were thus removed. In the retrieved 46 articles, 4 articles were additionally excluded according to the criteria. Total 42 articles were finally included: 19 RCTs, 15 non-RCTs (10 single-arm trials, 5 controlled trials), and 8 retrospective studies (Figure 1).
Thank you for your consideration of this manuscript.
